# Glucose-6-Phosphate Dehydrogenase (G6PD) Measurement Using Biosensors by Community-Based Village Malaria Workers and Hospital Laboratory Staff in Cambodia: A Quantitative Study

**DOI:** 10.3390/pathogens12030400

**Published:** 2023-03-01

**Authors:** Bipin Adhikari, Rupam Tripura, Lek Dysoley, Thomas J. Peto, James J. Callery, Chhoeun Heng, Thy Vanda, Ou Simvieng, Sarah Cassidy-Seyoum, Kamala Thriemer, Arjen M. Dondorp, Benedikt Ley, Lorenz von Seidlein

**Affiliations:** 1Mahidol-Oxford Tropical Medicine Research Unit, Faculty of Tropical Medicine, Mahidol University, Bangkok, Thailand; 2Centre for Tropical Medicine and Global Health, Nuffield Department of Clinical Medicine, University of Oxford, Oxford, UK; 3C.N.M National Center for Parasitology, Entomology and Malaria Control, Phnom Penh, Cambodia; 4Global and Tropical Health Division, Menzies School of Health Research and Charles Darwin University, Darwin, Australia

**Keywords:** village malaria worker, biosensor, vivax malaria, G6PD, primaquine, radical cure, Southeast Asia

## Abstract

Vivax malaria can relapse after an initial infection due to dormant liver stages of the parasite. Radical cure can prevent relapses but requires the measurement of glucose-6-phosphate dehydrogenase enzyme (G6PD) activity to identify G6PD-deficient patients at risk of drug-induced haemolysis. In the absence of reliable G6PD testing, vivax patients are denied radical curative treatment in many places, including rural Cambodia. A novel Biosensor, ‘G6PD Standard’ (SD Biosensor, Republic of Korea; Biosensor), can measure G6PD activity at the point of care. The objectives of this study were to compare the G6PD activity readings using Biosensors by village malaria workers (VMWs) and hospital-based laboratory technicians (LTs), and to compare the G6PD deficiency categorization recommended by the Biosensor manufacturer with categories derived from a locally estimated adjusted male median (AMM) in Kravanh district, Cambodia. Participants were enrolled between 2021 and 2022 in western Cambodia. Each of the 28 VMWs and 5 LTs received a Biosensor and standardized training on its use. The G6PD activities of febrile patients identified in the community were measured by VMWs; in a subset, a second reading was done by LTs. All participants were tested for malaria by rapid diagnostic test (RDT). The adjusted male median (AMM) was calculated from all RDT-negative participants and defined as 100% G6PD activity. VMWs measured activities in 1344 participants. Of that total, 1327 (98.7%) readings were included in the analysis, and 68 of these had a positive RDT result. We calculated 100% activity as 6.4 U/gHb (interquartile range: 4.5 to 7.8); 9.9% (124/1259) of RDT-negative participants had G6PD activities below 30%, 15.2% (191/1259) had activities between 30% and 70%, and 75.0% (944/1259) had activities greater than 70%. Repeat measurements among 114 participants showed a significant correlation of G6PD readings (r_s_ = 0.784, *p* < 0.001) between VMWs and LTs. Based on the manufacturer’s recommendations, 285 participants (21.5%) had less than 30% activity; however, based on the AMM, 132 participants (10.0%) had less than 30% activity. The G6PD measurements by VMWs and LTs were similar. With the provisions of training, supervision, and monitoring, VMWs could play an important role in the management of vivax malaria, which is critical for the rapid elimination of malaria regionally. Definitions of deficiency based on the manufacturer’s recommendations and the population-specific AMM differed significantly, which may warrant revision of these recommendations.

## 1. Introduction

### 1.1. Vivax malaria in Cambodia

Cambodia is rapidly reducing the transmission of malaria and plans to eliminate all human malarias by 2025 [1,2,3]. With a drop in *Plasmodium falciparum* cases in Cambodia, the true burden of vivax malaria becomes increasingly apparent. Almost 90% of all malaria cases in 2022 were due to *Plasmodium vivax* [4,5]. Vivax malaria can be challenging to treat due to the parasite’s dormant liver stage, hypnozoites, which can reactivate, cause relapse, and sustain ongoing transmission weeks to months after a primary infection [6]. To prevent relapses, radical cure regimens, which include 8-aminoquinolines, are critical to clear hypnozoites [7]. While 8-aminoquinoline-based radical cure is well-tolerated in most patients, it can cause drug-induced haemolysis in G6PD-deficient patients.

### 1.2. What Is G6PD Deficiency and Why Do We Need a Quantitative Measurement?

Glucose-6-phosphate dehydrogenase (G6PD) is a ubiquitous enzyme important to maintain the redox equilibrium in cells [8]. Low G6PD activities are collectively called G6PD deficiency and are the key risk factor for 8-aminoquinoline-induced haemolysis [9]. To date, more than 230 clinically relevant genetic variants have been described [10,11,12]. The underlying gene is located on the X chromosome; hence, men are either hemizygous G6PD-normal or G6PD-deficient. Women have two copies of the G6PD gene, one on each X chromosome, and are either homozygous G6PD-normal or G6PD-deficient, or heterozygous for the gene. A large proportion of heterozygous women have G6PD activities that yield a false-normal result when tested with qualitative diagnostics but are at risk of 8-aminoquinoline-induced haemolysis [13]. Heterozygous women can only be reliably identified using quantitative diagnostics. The gold standard, laboratory-based spectrophotometry, requires relatively sophisticated laboratory infrastructure and specifically trained technicians and is not suitable for point-of-care (PoC) diagnosis [14,15,16,17].

Routine G6PD testing is mostly done at the level of health centres, but people at risk of vivax malaria tend to live far from these facilities; consequently, only a fraction have access to routine testing [18]. Diagnostics that do not require a laboratory could address this gap. PoC diagnostics categorize patients according to G6PD activity and can guide the decision on the safest and most effective radical cure treatment regimen for individual patients [19,20]. Over the last years, several handheld diagnostics (biosensors) provide quantitative estimates of G6PD activity within minutes have been introduced. Biosensors were rolled out in Cambodia as PoC tests at primary and secondary healthcare facilities [18,21]. This strategy misses vivax malaria patients who are diagnosed by village malaria workers (VMWs) and unwilling or unable to travel to health facilities [22]. Providing G6PD diagnosis at the first point of contact, i.e., at the community level, can close this gap.

Using biosensors is more complex than using malaria rapid diagnostic tests (RDTs), and it is not known how well VMWs with their limited medical or laboratory training are able to perform these tests [18,21]. Operational evidence around the community use of biosensors is critical to direct future interventions and policies [18]. SD Biosensor (Republic of Korea) introduced the G6PD Standard (Biosensor), a mobile quantitative G6PD analyzer that can measure G6PD enzyme activity. In former field studies, the Biosensor showed good performance [23,24,25]. The Biosensor operates with a battery and measures G6PD activity and haemoglobin (Hb) levels by colorimetric methods from 10 µL of fresh blood within two minutes. Each measurement requires a single-use test strip and a test buffer and includes two pipetting steps with disposable blood-collecting devices (Ezi tubes) [23,26,27,28]. The primary objective of this study was to compare the G6PD activity measured by VMWs to activities measured by hospital-based LTs. The secondary objective of this study was to compare the G6PD deficiency categorization recommended by the manufacturer SD Biosensor to locally derived definitions.

## 2. Materials and Methods

### 2.1. Roll Out Radical Cure (RORC)

The Roll Out Radical Cure study was an operational study that examined multiple aspects of the feasibility of VMWs using Biosensors in the rural villages in Kravanh district, Pursat province, in western Cambodia [18,29].

VMWs ensure malaria diagnosis (based on RDT results), treatment, and referral of malaria patients and have limited access to laboratory-based tests. Most VMWs have only a few years of formal education, and their health education is focused on malaria diagnosis, treatment, and referral. VMWs receive a nominal incentive for each malaria case they treat and perhaps are mostly motivated by social recognition to offer community services on almost a voluntary basis. Most laboratory technicians (LTs) are specifically trained in laboratory sciences after secondary education and perform a wide spectrum of diagnostic tests covering multiple illnesses, including malaria. Prior to this study, LTs in Kravanh Referral Hospital were already using the Biosensor for an ongoing clinical trial and also as part of the deployment of Biosensors by the Cambodian healthcare system. Consistent with current practice, LTs were considered as a reference standard of G6PD measurement.

Between 2021 and 2022, a total of 28 VMWs and 5 LTs took part in the study. At the outset of the study, a one-day training workshop (T0) with both VMWs and LTs at Kravanh Referral Hospital was held. A second training workshop was held a month later (T30). Training details have been reported previously [18]. To avoid larger gatherings during the COVID-19 (coronavirus disease of 2019) pandemic, training sessions were adapted [30]. VMWs and LTs convened in smaller numbers (5 to 6 per event) for the workshop. Thus, multiple training workshops were held among the VMWs and LTs by training supervisors. The trainers also received regular training and supervision from experts. The training included standardized slides on how to use the Biosensor and practical sessions (testing each other).

### 2.2. Deployment of Biosensors in the Hands of Village Malaria Workers

Following the first workshop (T0), all VMWs and LTs were provided with Biosensors and supplies for 35 to 40 tests per month. It was assumed that VMWs would see around 30 RDT-negative febrile patients and 5 patients with an RDT-confirmed *P. vivax* infection per month. The VMWs collected written informed consent from each participant prior to study activities. After testing by the VMW, all vivax patients were referred for repeat testing and treatment at Kravanh Referral Hospital. Vivax patients were confirmed to have a parasite based on a microscopy. All patients with a positive *Plasmodium* test result and a subset of febrile patients (those willing to travel to the nearest health centre) were referred for a repeat G6PD testing by LTs (Figure 1). The LTs performed the repeat tests (using new samples but from same patients) at the research laboratory located within the district hospital using the Biosensor. Both VMWs and LTs recorded a brief clinical history of the patient and G6PD activity using a case record form (Appendix A).

VMWs convened each month at Kravanh Referral Hospital for a scheduled meeting with the district health department. At that time, VMWs also visited the adjacent study office to refill supplies, calibrate the Biosensor, and resolve queries. During the visit, VMWs also returned the case record forms (CRFs) that they had completed over the previous month. Throughout the study, all VMWs were in contact with the study coordinator via mobile phone to discuss concerns or issues.

### 2.3. Data Collection Instruments

The questionnaire (Appendix A) was designed and divided into various sections to collect anonymized socio-demographic data of (1) the VMWs and LTs, including age, sex, education, and years of experience (section-I); (2) the participants, including information on age, sex, education, and occupation (section-II); (3) the clinical history of participants (section-III); and (4) G6PD activity measurements (section IV). The questionnaire was designed, tested, and refined based on discussions with the VMWs and study coordinators.

### 2.4. Data Analysis

The Biosensor measures Hb levels within a range of 4 g/dL to 25 g/dL; however, G6PD readings can only be considered reliable if Hb readings are above 7 g/dL, based on the manufacturer’s recommendation. Furthermore, G6PD activity measurement in anaemic individuals can be spuriously increased [31]. G6PD readings with a corresponding Hb level below 7 g/dL or above 25 g/dL were therefore excluded from the analysis. The median activity of all paired readings by VMWs and LTs was calculated and compared.

All activities were categorized repeatedly. First, G6PD readings were categorized based on the population-specific AMM [19]. Since malaria may alter G6PD activity, the AMM was calculated excluding participants with confirmed *P. vivax* infection [32]. The AMM was defined as 100% G6PD activity. G6PD activities below 30% of the AMM were defined as deficient, activities at or above 30% and below 70% were defined as intermediate, and all other results were considered G6PD normal. Second, G6PD readings were classified based on recommendations provided by the manufacturer (<4.1 U/g Hb: deficient, 4.1 U/g Hb to <6.0 U/g Hb: intermediate, >6.0 U/g Hb: normal); and third, G6PD readings were categorized based on categories derived from an earlier field trial in Bangladesh (<2.6 U/g Hb: deficient, 2.6 U/g Hb to <6.0 U/g Hb: intermediate, >6.0 U/g Hb: normal) [33]. Paired readings by VMWs and LTs were categorized based on the AMM, and proportions were compared. The prevalence of G6PD deficiency was calculated based on the AMM of local study participants. Since G6PD deficiency may be protective against a *Plasmodium spp.* infection, participants with confirmed *Plasmodium* spp. infection were not considered when calculating G6PD deficiency prevalence [34,35].

The differences in proportions were calculated using the Chi-squared test, Fisher’s exact test, or the McNemar’s test for correlated proportions as appropriate. When needed, extended versions of each test were applied. Kappa was calculated to compare G6PD categories. Based on the data distribution, the two-sample Wilcoxon rank sum test and the Wilcoxon signed-rank test were employed to compare median activities measured by VMWs and LTs. Paired readings were further compared by calculating the Spearman correlation coefficient and by using Bland–Altman plots. The analysis was done in Stata version 14 (Stata Corp. 2015. Stata Statistical Software: Release 14. College Station, Texas, USA: StataCorp LP).

## 3. Results

### 3.1. Baseline

A total of 1275 RDT-negative febrile patients and 69 febrile patients with *P. vivax* infection were enrolled between May and September 2022. Haemoglobin (Hb) levels were below 7 g/dL or above 25 g/dL in 16 febrile patients and in 1 patient with *P. vivax* infection. Their G6PD readings were excluded from the analysis. A total of 1327 participants (98.7%) were included in the analysis. Duplicate readings were available from 114 participants (Figure 1).

A total of 33 Biosensor users participated in the study: 28 were VMWs, and 5 were LTs (Table 1). The median age of the Biosensor users was 42 years, and 51% (17/33) were women. The majority of LTs and VMWs had more than seven years of education (23/33; 70%) and more than six years of relevant experience (18/33; 55%).

A total of 1275 RDT-negative participants took part in the study. Out of 1275 total participants, 1235 were febrile during the recruitment (97%), with fever for a median duration of 2 days prior to presentation (Table 2). A total of 69 vivax patients participated in the study. Their median age was 22 years, 65/69 were men (94%), and 56/69 were farmers (81%). Most had a fever during recruitment (64/69; 93%), with a median duration of fever for 3 days. Most vivax patients (65/69; 94%) were men. Malaria RDT-negative participants were significantly older than malaria patients (*p* < 0.001), but neither G6PD activity nor Hb readings differed significantly between both cohorts (*p* > 0.05, Table 3).

### 3.2. Paired Results from VMWs and LTs

For 114 participants, two measurements were conducted—one by VMWs and another by LTs. Of that total, 18 readings were from female study participants (16%) and 96 readings from male participants (84%) (Figure 1). Median activities measured by VMWs (6.3 U/g Hb, interquartile range (IQR): 4.5–8.1) and LTs (6.7 U/g Hb, IQR: 5.2–7.5) did not differ significantly (*p* = 0.642), and both readings showed a positive, significant correlation (r_s_ = 0.784, *p* < 0.001; Figure 2 and Figure 3).

All paired readings from LTs and VMWs were categorized based on the AMM derived from the study population. The classification by VMWs and LTs was similar when considering the AMM (*p* = 0.070) and suggested a moderate but significant interrater agreement (κ = 0.588, *p* < 0.001) (Figure 3, Table 3).

### 3.3. Distribution of G6PD Activity

After removing measurements with Hb readings below 7 g/dl and above 25 g/dl, 1259 febrile and 68 vivax patients were retained in the statistical analysis. The AMM was 6.4 U/gHb (IQR: 4.5 to 7.8) based on 923 readings from RDT-negative (aparasitaemic) participants measured by VMWs (Table 4).

A total of 10% of the RDT-negative study population (n = 1259) had G6PD activities below 30% of the AMM, 15% of individuals had G6PD activities between 30% to 70%, and 75% had G6PD activities greater than 70%. Similar proportions were observed for vivax malaria (parasitaemic) participants (*p* = 0.340, Table 5 and Figure 4).

### 3.4. VMWs’ Interpretation of Biosensor Results

Readings of 1327 participants were categorized by VMWs, who considered 2.6 U/g Hb as the cut-off for deficient activities and 6.0 U/g Hb as the cut-off for intermediate activity (Table 4). In two cases, VMWs recorded “Error” rather than a category, and both readings were excluded, resulting in 1325 readings. In 24.5% of cases (n = 325), participants were categorized erroneously (*p* < 0.001). A total of 9 out of 1325 participants (0.7%) who were defined as G6PD-deficient by LTs were categorized as normal by VMWs. Overall, 6% (n = 76) of participants were categorized too high by the VMWs (e.g., a deficient individual was categorized as intermediate), and 22% (n = 294) of participants were categorized too low (Table 6).

### 3.5. Comparing G6PD Categories: AMM-Based vs. Manufacturer-Recommended

In a paired comparison, the proportions differed significantly between the manufacturer’s recommendations and the classification based on the site-specific AMM (*p* < 0.001); 87% of individuals (n = 285) classified as intermediate according to the manufacturer’s recommendation had more than 70% activity (median activity: 85.9%, IQR: 79.7–90.6), and 54% (n = 153) had more than 30% activity (median activity: 45.3%, IQR: 35.9–54.7). (Table 4 and Table 7, Figure 1).

## 4. Discussion

This study assessed VMWs’ G6PD measurements using Biosensors and, in a subset of samples, compared VMWs’ G6PD readings with those of LTs. There are two main findings: Firstly, VMWs’ G6PD readings in the community were similar to those of trained laboratory technicians. Secondly, a comparison between G6PD categories based on the locally calculated AMM and the manufacturer’s recommendations showed a significant difference.

### 4.1. VMWs’ Competence in Using the Biosensor and Implications

VMWs’ competence in using the Biosensor was high in the analysis of paired results between VMWs and LTs—suggesting that the use of Biosensors at the community level is feasible. A major consideration for a PoC community management of vivax malaria is the need for training, supervision, and monitoring of G6PD test use, and specifically supporting VMWs in G6PD categorization and corresponding treatment [18]. The findings of this study increase optimism for shifting vivax malaria management beyond health centres and toward the initial point of contact [22]. Previous reports have highlighted multiple barriers for remote residents to access health centres for vivax malaria treatment. A recent study reported the beliefs held by health authorities and health staff that VMWs may not be qualified to perform G6PD testing [36]; and thus, this warrants more operational studies to forge the evidence. Since the travel to health centres and adherence to radical cure treatment are major barriers in the management of vivax malaria, providing essential resources at a community level would outweigh the costs of investment [18,22]. A community health worker (CHW)-led integrated community management of malaria in Uganda was found to be cost-efficient when accounting for both direct medical costs and costs of transport in relation to care-seeking and was most appropriate in terms of treatment when compared with the alternative options [37]. Globally, the evidence around the benefits (cost, coverage, quality of services, and reductions in disease morbidity and mortality) of community management of malaria, including other illnesses, by CHWs are growing [38,39,40,41,42]. In regions where vivax malaria transmission continues only in remote and inaccessible areas, community management is the most promising avenue for rapid malaria elimination [18,22,43].

Such an approach requires additional efforts and resourcing. Study coordinators were on standby with mobile phones to address any concerns from the VMWs. This allowed VMWs to resolve queries in real-time. VMWs were also provided with smart phones to record G6PD and Hb measurements by taking a picture of results, helping study coordinators resolve concerns. With the increasing digitalization of health services, VMWs based in remote villages in Cambodia can be safely supervised and monitored through the use of telephones and other digital resources [44].

### 4.2. G6PD Prevalence and Categorization

VMWs’ measurement of G6PD showed high consistency with the LTs; nonetheless, the interpretation and corresponding treatment require further support. Out of 1325 participants, 191 were G6PD-deficient based on definitions from a study in Bangladesh [33]. VMWs inaccurately categorized nine of these G6PD-deficient participants as G6PD-normal. Similarly, of the 394 intermediate participants, VMWs miscategorized 16% (61/394) as normal. Such a miscategorization would mean that women categorized as intermediate with reduced G6PD activity may have a haemolytic reaction when exposed to 8-aminoquinolines. The overall risk of this miscategorization for the cohort of 1325 participants is less than 5% (61/1325). In this study, training was focused primarily on the correct use of the Biosensor—a critical first step for the use of Biosensors by VMWs. In contrast, less emphasis was placed on the interpretation of results and treatment. Specifically, the VMWs were not provided with a job aid that would have guided and prompted appropriate interpretation of G6PD test results. Similar to decision support tools used by clinicians, such as a ‘job aid’ with laboratory values and their interpretations, providing reference values for G6PD categories and corresponding treatments would likely improve the interpretation [45,46,47]. The Cambodian national malaria control programme already provides a comprehensive digital application (mobile app) to healthcare workers—a support for malaria case identification, referral, follow-up, and recording [22].

In this study, 10% of malaria RDT-negative participants had G6PD activity below 30% when considering the site-specific AMM, which is consistent with previous estimates of the prevalence of G6PD deficiency in Cambodia that range from 7 to 13% [48,49]. This study showed a significant discrepancy between the manufacturer’s recommended categories and the categories based on the locally calculated AMM. The manufacturer-recommended cut-off is likely to be too conservative in this specific context and could lead to the unnecessary exclusion of eligible patients from receiving radical cure treatment, given the Biosensor’s reported good reproducibility [50].

### 4.3. Strengths and Limitations

This study was integrated into the VMWs’ regular schedule in Kravanh district, and thus using Biosensors, recording the readings, and completing CRFs were additional responsibilities amongst their competing priorities. Unlike the evidence around the use of Biosensors among laboratory-based health workers, this study builds evidence from an operational standpoint, where G6PD testing was integrated in the VMWs’ routine work and thus highlights pragmatic issues. Although a large number of participants were enrolled (n = 1344), only 114 had duplicate readings for the comparative analysis. In this study, G6PD categorization was based on previous training materials used for Bangladesh, and thus the VMWs’ classification differed from both the manufacturer’s categories and the categories based on the local AMM. The study cannot attribute the competence and output of VMWs on using the Biosensor based on the training alone, because myriad other factors can contribute to their skills and outcome. Additionally, it would have been of interest to compare Biosensor readings with the gold standard, spectrophotometry. Despite repeated attempts, it was not possible to set up spectrophotometry at the study site during the study period, which coincided with the COVID-19 pandemic.

## 5. Conclusions

VMWs measured G6PD activity in the community with the support of training, supervision, and monitoring. The readings between VMWs and LTs were similar. In the future, VMWs can be resourced and trained to measure G6PD activity and provide vivax malaria treatment in the communities under supervision. In remote and inaccessible communities, resourcing VMWs could be the best alternative to overcome the barriers that currently prevent patients from receiving radical cure at health centres.

The manufacturer-recommended cut-offs for determining G6PD deficiency did not match the population-specific cut-offs and could result in the under-treatment of a significant proportion of patients eligible for safe radical cure regimens. The appropriate use of Biosensors at the point of first contact with the patient could be the critical step for malaria elimination in vivax endemic regions.

## Figures and Tables

**Figure 1 pathogens-12-00400-f001:**
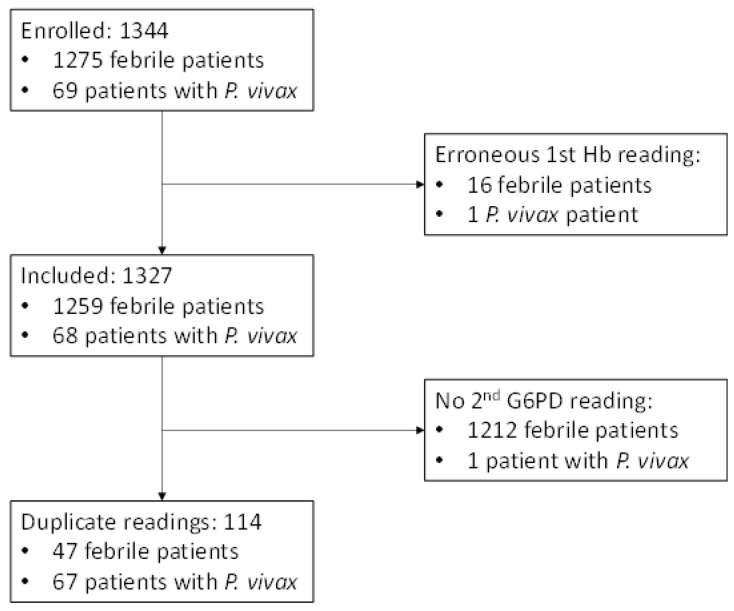
Consort chart.

**Figure 2 pathogens-12-00400-f002:**
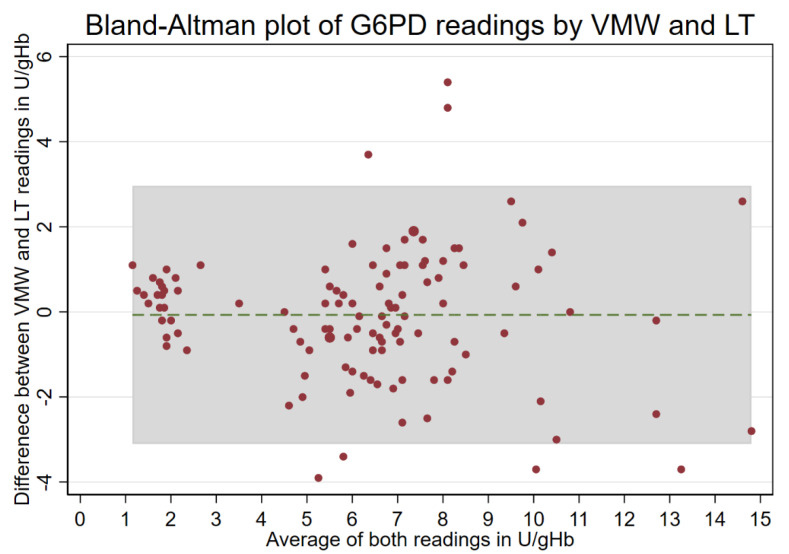
Paired G6PD readings by VMWs and LTs. Red dots: G6PD values, green dotted line: mean difference, grey shaded area: 95% limits of agreement.

**Figure 3 pathogens-12-00400-f003:**
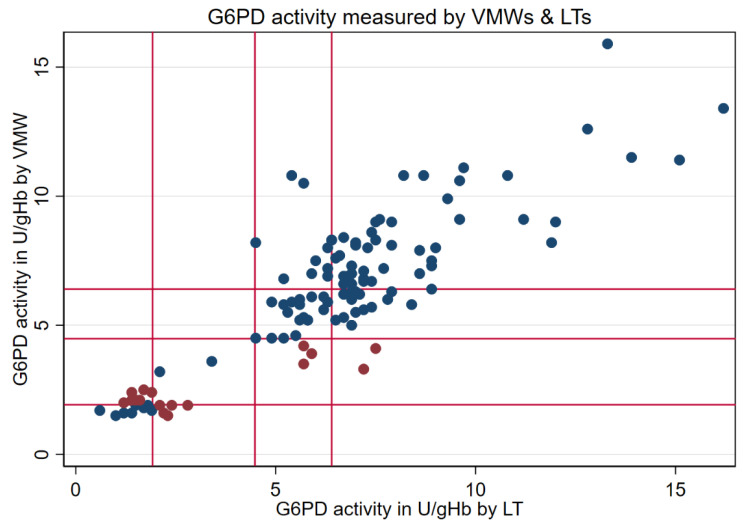
G6PD activity measured by VMWs and LTs. Vertical lines from origin outwards: 30%, 70%, and 100% activity, respectively; blue dots: consistent categorization; red dots: inconsistent categorization; VMW: village malaria worker; LT: laboratory technician.

**Figure 4 pathogens-12-00400-f004:**
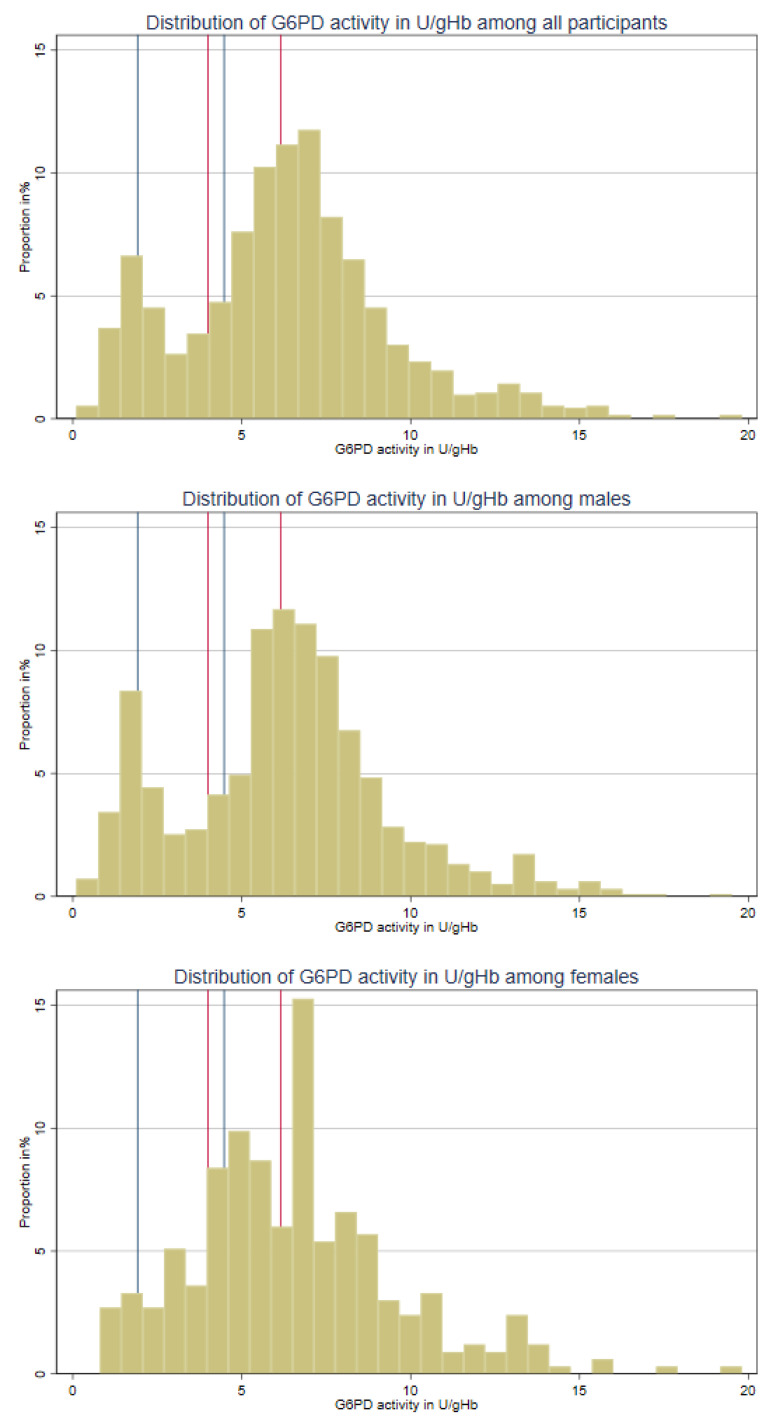
G6PD activity distribution based on AMM among RDT-negative and parasitaemic participants (AMM: adjusted male median; RDT: rapid diagnostic test). Blue lines are based on AMM of study participants: 30% and 70% activity, respectively. Red lines are based on the package insert: 4.0 U/g Hb and 6.1 U/g Hb, respectively.

**Table 1 pathogens-12-00400-t001:** Socio-demographics of Biosensor users stratified by professional background (n = 33).

Characteristics	LTs (n = 5)	VMWs (n = 28)	
	Number (%)	Number (%)	*p*-Value
Age group	Median = 42 years (IQR = 36–51.5)		
≤40 years	2 (40)	9 (32.1)	0.55
≥41 years	3 (60)	19 (67.9)	
Sex			
Female	1 (20)	16 (57.1)	0.02
Male	4 (80)	12 (42.9)	
Education	Median = 7 years (IQR = 6–9)		
≤6 years	1 (20)	9 (32.1)	0.51
≥7 years	4 (80)	19 (67.9)	
Experience	Median = 6 years (IQR = 3–6)		
≤5 years	2 (40)	13 (46.4)	0.5
≥6 years	3 (60)	15 (53.6)	

IQR: interquartile range; VMW: village malaria worker; LT: laboratory technician.

**Table 2 pathogens-12-00400-t002:** Characteristics of Malaria RDT-negative febrile and vivax malaria patients.

Characteristics of Participants	RDT-Negative Patients (n = 1275)	Vivax Patients (n = 69)	
	Number (%)	Number (%)	*p*-Value
Age (Years)			
Median (IQR)	32 (24–42)	22 (18–33)	<0.001
Sex			
Male	937 (73.5)	65 (94.2)	
Female	338 (26.5)	4 (5.8)	<0.001
Education in years			
Median (IQR)	5 (7–3)	5 (7–3)	0.570
Occupation			
Farmer	1199 (94)	56 (81.2)	<0.001
Other	76 (6)	13 (18.8)	
Do you currently have fever?			
Yes *	1235 (96.9)	64 (92.8)	
No	37 (2.9)	5 (7.2)	
Other	3 (0.2)	0 (0.0)	0.150
Fever in days			
Median (IQR)	2 (2–3)	3 (2–3)	<0.001
Have you suffered from malaria in the past (entire lifetime)?			
Yes **	848 (66.5)	51 (73.9)	
No	427 (33.5)	18 (26.1)	0.230
If you had malaria in the past what kind of malaria was it?			
Falciparum	181 (22)	1 (2)	
Vivax	643 (78)	49 (98)	<0.001

* sub-sample with fever; ** sub-sample who had malaria in the past and could recall. IQR: interquartile range. RDT: rapid diagnostic test.

**Table 3 pathogens-12-00400-t003:** Categorization by VMWs and LTs, considering the AMM as 100% activity.

LTs					
	G6PD Category (AMM%)	Normal	Intermediate	Deficient	Total
VMWs	Normal (>70%)	87 (75.7)	0 (0.0)	0 (0.0)	87 (76.3)
	Intermediate (30 to 70%)	5 (4.4)	2 (1.7)	7 (6.1)	14 (12.3)
	Deficient (≤30%)	0 (0.0)	5 (4.4)	8 (7.0)	13 (11.4)
	Total	92 (80.7)	7 (6.1)	15 (13.2)	114 (100.2)

VMW: village malaria worker; LT: laboratory technician; AMM: adjusted male median.

**Table 4 pathogens-12-00400-t004:** Comparing alternative classification schemes with the locally established AMM.

G6PD Category	Based on AMM (U/g Hb)	Based on Field Implementation * (U/g Hb)	Based on Manufacturer’s Recommendation (U/g Hb)
Deficient (≤30%)	≤1.9	≤2.6	≤4.0
Intermediate (30% to 70%)	>1.9 to 4.5	>2.6 to 6.0	>4.0 to 6.0
Normal (>70%)	>4.5	>6.0	>6.0

* Based on the definitions from a previous field trial conducted in Bangladesh [33]. AMM: adjusted male median.

**Table 5 pathogens-12-00400-t005:** Distribution of G6PD categories based on AMM among parasitaemic and RDT-negative participants.

Category	Men (%)	Women (%)	Total (%)
RDT-negative patients			
≥70%	697 (75.0)	247 (74.9)	944 (75.0)
≥30% to <70%	124 (13.4)	67 (20.3)	191 (15.2)
<30%	108 (11.6)	16 (4.9)	124 (9.9)
Total	929 (100.0)	330 (100.0)	1259 (100.0)
Parasitaemic (Vivax patients)			
≥70%	51 (79.7)	3 (75.0)	54 (79.4)
≥30% to <70%	6 (9.4)	0 (0.0)	6 (8.8)
<30%	7 (10.9)	1 (25.0)	8 (11.8)
Total	64 (100.0)	4 (100.0)	68 (100.0)

AMM: adjusted male median; RDT: rapid diagnostic test.

**Table 6 pathogens-12-00400-t006:** G6PD readings categorized by VMWs and calculated categories.

		Categorization by VMWs			
	Category	Normal	Intermediate	Deficient	Total (%)
Calculated G6PD category *	Normal (%)	607 (82.0)	102 (13.8)	31 (4.2)	740 (55.9)
	Intermediate (%)	61 (15.5)	217 (55.1)	116 (29.4)	394 (29.7)
	Deficient (%)	9 (4.7)	6 (3.1)	176 (92.2)	191 (14.4)
	Total (%)	677 (51.1)	325 (24.5)	323 (24.4)	1325 (100.0)

* Based on AMM from a previous field trial conducted in Bangladesh [33]. VMWs: village malaria workers.

**Table 7 pathogens-12-00400-t007:** Agreement between classification schemes: manufacturer’s recommendation vs. AMM-based % activity.

	G6PD Category Based on Local AMM				
G6PD category based on manufacturer’s recommendation	Category	Normal	Intermediate	Deficient	Total (%)
	Normal (%)	713 (100.0)	0 (0.0)	0 (0.0)	713 (53.7)
	Intermediate (%)	285 (86.6)	44 (13.4)	0 (0.0)	329 (24.8)
	Deficient (%)	0 (0.0)	153 (53.7)	132 (46.3)	285 (21.5)
	Total (%)	998 (75.2)	197 (14.4)	132 (10.0)	1327 (100.0)

AMM: adjusted male median.

## Data Availability

The data is available upon request to the Mahidol Oxford Tropical Medicine Research Unit Data Access Committee. Available online at http://bit.ly/3IBXKu5 (accessed on 28 February 2023) complying with the data access policy.

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
