# Peer review of "Glucose-6-Phosphate Dehydrogenase (G6PD) Measurement Using Biosensors by Community-Based Village Malaria Workers and Hospital Laboratory Staff in Cambodia: A Quantitative Study"

_pathogens, 2023, doi:10.3390/pathogens12030400_

Round 1

Reviewer 1 Report

Vivax malaria poses a public health threat in communities that leave endemic areas. As such, in rural communities, the difficulty to access healthcare facilities can amplify the recursive infection and maintain the disease in pre-elimination areas. Moreover, because the parasite can remain dormant for several years it creates a reservoir of careers asymptomatic. The development of a device capable of estimating G6DP activity becomes very useful, especially if it could be used by village malaria workers who happen to have a low lab education. The subject discussed by the authors is very important and accessing the device accurately could enormously contribute to reducing the human reservoir of Plasmodium vivax. As such, the authors have access to both the accuracy of the device reading and the classification of the malaria-vivax-infected population. However, there are few issues to address to make the core message clear to readers. In the following lines, we will discuss the issues we found that could be addressed in the abstract, the introduction, the methods, the results, and the discussions. Given the importance of the topic, we will ask questions and make some suggestions. The answer to the questions will clarify the storyline of the manuscript.

Abstract

1-     We suggest to re-adjust the goal of the manuscript in the abstract since the main goal is written “ assess whether community-based village malaria workers (VMWs) can accurately use the biosensor by comparing G6PD activity readings by VMWs with readings of hospital-based laboratory technicians (LTs) in Kravanh district, Cambodia” does not match with what the authors do in the main text which is accessing the biosensor device perform/ accuracy.

2-     On line 31 it was mentioned '...patient biosensors', are all patients provided with biosensors? or is the device at the disposal of the community?

3-     On the same line, do the VMW measure the G6PD, or do they estimate it? (If it is an estimation, please, in the method section, describe how the estimation was done) (the device estimates and VMW read)

4-     Line 33: How is this sample subset selected? How are false negative patients handled?

5-     Line 35: Why is the adjusted male median metric calculated?

6-     Line 44: The authors mentioned a significant correlation of results, how is the correlation measured? (What are the variables that are correlated?)

7-     Lines 45-47: the authors imply that there is a site-dependent variability observed in the classification. How is this difference accessed? Why is the difference observed? How does this discrepancy use to access the VMWs’ skills acquisition? What are the implications of this discrepancy? Are the VMW classifications better than the manufacturer's recommendation?

Introduction

1-     Line 67: No need to specify the number in the brackets

2-     Line 70: Is 8-aminoquinolines the only remedy used to treat P vivax? If not, please include the sentence that informs us about why you are putting an emphasis on 8-aminoquinolines in the geographical context.

3-     Line 91: Define the acronym PoC

4-     Lines 109-114: The first objective formulated as the way is that it does not match with the goal of the abstract. The formulation looks more like an evaluation of the performance of the device, and the performance of the VMW is more like a byproduct of this evaluation. This section implies more accessing the device performance, and this needs to be revised in the abstract.

Materials and Methods

1-     Inform the reader about why you are accessing the performance of the device. What are the estimated sensitivity and specificity of the device?

2-     Lines 138-139: Define the acronyms before using them

3-     Lines 147-150: Are there some false positives in the sample?

4-     Lines 160-163: the use of ‘tester’ is confusing, are the authors referring to VMW if yes, please use VMW instead. Why is personally identifiable information from the patients and VMW collected? How are they used in the context of the study? (Questions 2 and 3)

5-     Lines 167-169: Does it mean that there were no G6PG in the eliminated range and how is this information relevant to accessing the performance of both VMW and biosensor devices?

6-     Line 171: the authors need to explain the subset they are referring to because the reading happens after the ‘diagnostic’ of VMW (read of the device or fever and acceptance to travel) now a new subset sounds to be defined based on the read. They can present a table reflecting the sample size finally used to reach their goal (could, for instance, move figure 1 below this line)

7-     Lines 183-186: The authors compare the reading of the devices with the reading from the lab. How are the time laps between the two readings adjusted in the comparison? In case there is no time gap between the two readings, I suggest using a paired t-test instead.    

8-     Lines 188-194: The authors need to be more specific about the statistical analysis performed, the variable used, and the rationale for their choice.

Results

1-     Lines 209-210: What is the rationale behind splitting the sample and why is this choice not explained in the method section? What statistical test is performed to obtain the p-value?

2-     Why is an emphasis placed on patients, not on the device or the worker?

3-     Lines 219-221: Why do the authors elaborate on the anamnesis of patients?  How is this relevant to the overarching goal of the study?

4-     I suggest replacing Figures 2 and 3 with a paired t-test, as it is statistically more powerful than the Bland-Altman visualisation method used. In addition, there is a risk of increasing type I error by combining the Bland-Altman and the correlation test. Furthermore, the correlation is performed on the mean not on the median since Pearson's correlation is a parametric test.

5-     Table 3 is not standing alone; how is the categorisation of VMW different from the LTs?

6-     Line 243: What statistical analysis is used?

7-     Table 4: remove the second column or merge it with the first column, put the unit below the name of the column, and keep the numbers only in the table.

8-     Table 5: Why are the authors divided the sample according to gender? They do not need to include the row and column total on the table and they can replace categories by their denomination

9-     Fig. 4: What is the information that the reader needs to have here? I suggest removing the plot showing the distribution.

10-   Lines 261-262: need to come before Table 4. How are the p-values obtained?

11-   Lines 264-282: to prevent type I error I suggest running either a chi-square or a fisher exact test on a table that will be like this

LTs

Local

Manufacturer

Normal

Intermediate

Deficient

Where each cell contains a counting number.  

Discussion

1-     Lines 295-295: it is hard given the results presented to associate the discrepancy between observed count and predicted count to associate that to a lack of training since intrinsically the study does not have some checking point on wrongdoing or biased from VMW

Author Response

Date: 12th February 2023

Dear reviewers,

Thank you very much for your constructive feedback and opportunity to revise our work. We have utilized your suggestions to improve our manuscript. Please see below our responses to your specific suggestions with corresponding changes in the manuscript.

We look forward to your kind consideration.

Sincerely,

Bipin Adhikari

Reviewer #1

Vivax malaria poses a public health threat in communities that leave endemic areas. As such, in rural communities, the difficulty to access healthcare facilities can amplify the recursive infection and maintain the disease in pre-elimination areas. Moreover, because the parasite can remain dormant for several years it creates a reservoir of careers asymptomatic. The development of a device capable of estimating G6DP activity becomes very useful, especially if it could be used by village malaria workers who happen to have a low lab education. The subject discussed by the authors is very important and accessing the device accurately could enormously contribute to reducing the human reservoir of Plasmodium vivax. As such, the authors have access to both the accuracy of the device reading and the classification of the malaria-vivax-infected population. However, there are few issues to address to make the core message clear to readers. In the following lines, we will discuss the issues we found that could be addressed in the abstract, the introduction, the methods, the results, and the discussions. Given the importance of the topic, we will ask questions and make some suggestions. The answer to the questions will clarify the storyline of the manuscript.

Abstract

  • We suggest to re-adjust the goal of the manuscript in the abstract since the main goal is written “ assess whether community-based village malaria workers (VMWs) can accurately use the biosensor by comparing G6PD activity readings by VMWs with readings of hospital-based laboratory technicians (LTs) in Kravanh district, Cambodia” does not match with what the authors do in the main text which is accessing the biosensor device perform/ accuracy.

AUTHORS: We have revised to add the specific objectives to align with the methods and results in the study and the revised objectives read as follows:

The objectives of this study were to 1. compare the G6PD activity readings using biosensor by Village malaria workers (VMWs) and hospital-based laboratory technicians (LTs); and 2. compare the G6PD deficiency categorization recommended by the biosensor manufacturer with categories derived from a locally estimated adjusted male median (AMM) in Kravanh district, Cambodia.

  • On line 31 it was mentioned '...patient biosensors', are all patients provided with biosensors? or is the device at the disposal of the community?

AUTHORS: Apologies, the statement in line 31 does not mention ‘patient biosensors’, but patients using biosensors’. No, patients were not provided biosensors, the device was only distributed to VMWs, and only VMWs used it. We have changed the respective section which now reads as follows:

Participants were enrolled between 2021 and 2022 in western Cambodia. 28 VMWs  and five LTs received standardized training on the Biosensor and were provided with a machine each. G6PD activities of febrile patients identified in the community were measured by VMW; in a subset a second reading was done by LTs. All participants were tested for malaria by rapid diagnostic test (RDT). The adjusted male median (AMM) was calculated from all RDT negative participants and defined as 100% G6PD activity.

  • On the same line, do the VMW measure the G6PD, or do they estimate it? (If it is an estimation, please, in the method section, describe how the estimation was done) (the device estimates and VMW read)

AUTHORS: We have revised the section and reads as follows:

Participants were enrolled between 2021 and 2022 in western Cambodia. 28 VMWs and five LTs received standardized training on the Biosensor and were provided with a machine each. G6PD activities of febrile patients identified in the community were measured by VMW; in a subset a second reading was done by LTs. All participants were tested for malaria by rapid diagnostic test (RDT). The adjusted male median (AMM) was calculated from all RDT negative participants and defined as 100% G6PD activity.

  • Line 33: How is this sample subset selected? How are false negative patients handled?

AUTHORS: All vivax patients were referred to the Kravanh district hospital for vivax malaria treatment as a part of routine management of vivax malaria. In Cambodia, VMWs use RDT to diagnose vivax malaria patients in the community, if they are RDT positive, they are referred to health centre or district hospital for radical cure treatment. If they are RDT-negative, but febrile, depending upon the severity of illnesses and willingness to travel, they are also referred to health centre/hospital for further exploration of disease diagnosis. There are no specific measures for RDT negative (false negative patients) patients. In this study, a subset of febrile patients was asked if they would like to know more about the causes of fever they were suffering. Based on their willingness to travel, they were referred to hospital where they were repeat tested for G6PD by LTs. The methodology of recruitment has been elaborated in the main text.

  • Line 35: Why is the adjusted male median metric calculated?

AUTHORS: The adjusted male median is a population specific metric to define G6PD deficiency. This is clarified in the following sentence and supported by the original article by Domingo et al (2013), in second paragraph under the section ‘Data analysis’:First, G6PD readings were categorized based on the population-specific AMM [19].

  • Line 44: The authors mentioned a significant correlation of results, how is the correlation measured? (What are the variables that are correlated?)

AUTHORS: The correlation of G6PD readings between VMWs and LTs were assessed using spearman correlation coefficient (rs). The statement has been revised and reads as follows:

Repeat measurements among 114 participants showed a significant correlation of G6PD readings (rs=0.784, p<0.001) between VMWs and LTs.

  • Lines 45-47: the authors imply that there is a site-dependent variability observed in the classification. How is this difference accessed? Why is the difference observed? How does this discrepancy use to access the VMWs’ skills acquisition? What are the implications of this discrepancy? Are the VMW classifications better than the manufacturer's recommendation?

AUTHORS: We have assessed the difference in the population specific classifications and the manufacturer recommended classification in table 7 where we quantify the proportion of individuals with discrepant categories. We have clarified table 7 in the corresponding paragraph under the heading ‘Comparing G6PD categories: AMM-based vs manufacturer recommended’ where it now reads (Lines 268-274):

In a paired comparison, proportions differed significantly between the manufacturer recommendations and the classification based on the site-specific AMM (p<0.001); 87% of individuals (n=285) classified as intermediate according to manufacturer recommendation had more than 70% activity (median activity: 85.9%, IQR: 79.7 – 90.6), and 54% (n=153) had more than 30% activity (median activity: 45.3%, IQR: 35.9 – 54.7).

We do not discuss why local definitions of G6PD deficiency vary between populations as we did not assess this. We believe this is due to difference in G6PD deficiency prevalence and differences in the proficiency of end users.

The observed differences do not impact on the ability of VMWs to categorise Biosensor results. Since we did not include the gold standard spectrophotometry (lines 351 – 353), we cannot tell which definition is more suitable but only state that the manufacturer recommendations are more conservative: “Thirdly, a comparison between G6PD categories based on the locally calculated AMM and the manufacturers recommendations showed a significant difference.” (lines 277 – 279)  

Introduction

  • Line 67: No need to specify the number in the brackets

AUTHORS: We have removed number and the brackets. 

  • Line 70: Is 8-aminoquinolines the only remedy used to treatP vivax? If not, please include the sentence that informs us about why you are putting an emphasis on 8-aminoquinolines in the geographical context.

AUTHORS: Yes, 8-aminoquinoloesare are the only licensed group of drugs that can clear the hypnozoites from the human host (Line 58- 61).

  • Line 91: Define the acronym PoC

AUTHORS: Thank you for the suggestion. Acronym PoC is defined in the last statement of the preceding paragraph (Line 75).

  • Lines 109-114: The first objective formulated as the way is that it does not match with the goal of the abstract. The formulation looks more like an evaluation of the performance of the device, and the performance of the VMW is more like a byproduct of this evaluation. This section implies more accessing the device performance, and this needs to be revised in the abstract.

AUTHORS: Thank you for the suggestion. We have revised the objectives in the abstract and in the main text as well.

The primary objective of this study was to compare the G6PD activity measured by VMWs to activities measured by hospital based LTs. The secondary objective of this study was to compare the G6PD deficiency categorization recommended by the manufacturer SD Biosensor to locally derived definitions. (Line 98-101)

Materials and Methods

  • Inform the reader about why you are accessing the performance of the device. What are the estimated sensitivity and specificity of the device?

AUTHORS: We have revised and added studies assessing sensitivity and specificity in the last paragraph of the introduction section and reads as follows:

SD Biosensor (Republic of Korea) introduced the G6PD Standard (Biosensor), a mobile quantitative G6PD analyzer that can measure G6PD enzyme activity. In former field studies the Biosensor showed good performance [23-25].

  • Lines 138-139: Define the acronyms before using them

AUTHORS: The acronyms are deleted.

  • Lines 147-150: Are there some false positives in the sample?

AUTHORS: Thank you for the question. We diagnosed vivax patients based on the RDT by VMWs in the village. The patients were re-examined by microscopy for a different study (not within the scope of this study) and all RDT positive patients were confirmed to have Plasmodium vivax parasite. The revised statement reads as follows:

All vivax patients were confirmed to have parasite based on the microscopy (Line 135-136)

  • Lines 160-163: the use of ‘tester’ is confusing, are the authors referring to VMW if yes, please use VMW instead. Why is personally identifiable information from the patients and VMW collected? How are they used in the context of the study? (Questions 2 and 3)

AUTHORS: Thank you for the question. We have revised the statement and replaced the word ‘testers’. We collected socio-demographics of VMWs and LTs to present the descriptive characteristics of the participants (all information is adequately anonymized and are not personally identifiable). Its critical to know who the participants are, their background socio-demographics and illness history, especially as a community-based study for follow-up and support that was crucial in the study. 

The revised text reads as follows:

Data collection instruments

The questionnaire (Appendix A) was designed to collect anonymized socio-demographic data of 1) the VMWs and LTs, including age, sex, education, and years of experience (section-I); 2) the participants, including information on age, sex, education, and occupation (section-II); 3) the clinical history of participants (section-III); and 4) G6PD activity measurements (section IV). The questionnaire was designed, tested, and refined based on discussions with the VMWs and study coordinators. 

  • Lines 167-169: Does it mean that there were no G6PD in the eliminated range and how is this information relevant to accessing the performance of both VMW and biosensor devices?

AUTHORS: This only means the G6PD estimates can be unreliable when a person’s Hemoglobin measures are outside the range of 4 to 25g/dL. It did not affect the performance of VMWs or LTs but biosensor’s estimation of G6PD could be unreliable. So, the G6PD estimates from extreme Hemoglobin measurements were excluded.

  • Line 171: the authors need to explain the subset they are referring to because the reading happens after the ‘diagnostic’ of VMW (read of the device or fever and acceptance to travel) now a new subset sounds to be defined based on the read. They can present a table reflecting the sample size finally used to reach their goal (could, for instance, move figure 1 below this line)

AUTHORS: We have revised the paragraph to avoid the potential lack of clarity. We have cited the figure 1 as well to add clarity. The revised paragraph reads as follows:

All patients with a positive Plasmodium test result and a subset of febrile patients (those willing to travel to the nearest health centre) were referred for a repeat G6PD testing by LTs (Figure 1). Paired median of G6PD readings by VMWs and LTs were compared. Lines: 136-138 and 161

  • Lines 183-186: The authors compare the reading of the devices with the reading from the lab. How are the time laps between the two readings adjusted in the comparison? In case there is no time gap between the two readings, I suggest using a paired t-test instead.    

AUTHORS: As the reviewer has highlighted, there was a delay between testing by VMWs and LTs. Since testing was done from a fresh sample collected from the same participant we do not think that this delay will have resulted in a significant difference in G6PD activities. This is highlighted in the sentence:

LTs performed the repeat-tests (using new samples but from same patients) at the research laboratory located within the district hospital using the Biosensor. (138 – 140)

Since G6PD measurements were not normally distributed we used the Wilcoxon signed rank test for paired measurements. This is highlighted in the sentence as:

Based on the data distribution, the two sample Wilcoxon rank sum test and the Wilcoxon signed rank test were employed to compare median activities measured by VMW s and LTs. (lines 180 – 182)

  • Lines 188-194: The authors need to be more specific about the statistical analysis performed, the variable used, and the rationale for their choice.

AUTHORS: We have revised the respective paragraph that now reads:

Differences in proportions were calculated using the Chi-squared test, Fisher’s exact test or the McNemar’s test for correlated proportions as appropriate. When needed extended versions of each test were applied. Kappa was calculated to compare agreement of G6PD categories. Based on the data distribution, the two sample Wilcoxon rank sum test and the Wilcoxon signed rank test were employed to compare median activities measured by VMW s and LTs. Paired readings were further compared by calculating the Spearman correlation coefficient and using Bland-Altman plots. Analysis was done in Stata version 14 (Stata Corp, USA). (lines 176 – 183)

Results

  • Lines 209-210: What is the rationale behind splitting the sample and why is this choice not explained in the method section? What statistical test is performed to obtain the p-value?

AUTHORS: Table 1 describes characteristics of VMWs and LTs. To clarify this, we have changed the heading of the table to: “Socio-demographics of biosensor users stratified by professional background (n=33).” (line 209). The test applied was a Fishers exact or Chi squared test as appropriate and described in the methods section (kindly see above).

Why is an emphasis placed on patients, not on the device or the worker?

AUTHORS: The table 2 presents descriptive characteristics of the participants to illustrate how many participants we had and their socio-demographic and brief clinical features. Rest of the tables and figures are main findings, and they are emphasized adequately.

  • Lines 219-221: Why do the authors elaborate on the anamnesis of patients?  How is this relevant to the overarching goal of the study?

AUTHORS: As described above, these are the patients who participated in this study, and while presenting comparative G6PD estimates of these participants it is tempting for readers to understand their socio-demographics and clinical characteristics.

  • I suggest replacing Figures 2 and 3 with a paired t-test, as it is statistically more powerful than the Bland-Altman visualisation method used. In addition, there is a risk of increasing type I error by combining the Bland-Altman and the correlation test. Furthermore, the correlation is performed on the mean not on the median since Pearson's correlation is a parametric test.

AUTHORS: Figure 2 is a Bland-Altman plot and we would like to include this as we think this is a good way to display measurements of VMWs and LTs and show their relationship. The respective analysis is purely descriptive and does not increase the risk of a type I error. Figure 3 is quite similar to a Bland-Altman analysis, however, allows to specifically highlight contradictory readings which we find quite intuitive. Due to its descriptive nature, we don’t see the risk of a type I error. The reviewer suggests a paired t-test, which is not suitable due to the nature of the data. We have compared readings of VMWs and LTs using the equivalent tests for non-parametric data, the Wilcoxon signed rank test as is described in lines 180 – 184 “Based on the data distribution, the two sample Wilcoxon rank sum test and the Wilcoxon signed rank test were employed to compare median activities” and lines 217 – 219 where it says: “Median activities measured by VMWs (6.3U/g Hb, Interquartile range (IQR): 4.5 – 8.1) and LTs (6.7U/g Hb, IQR: 5.2 – 7.5) did not differ significantly (p=0.642)…”

Table 3 is not standing alone; how is the categorisation of VMW different from the LTs?

AUTHORS: Table 3 is a 2x2 table that compares paired and categorized readings of VMWs and LTs based on the AMM. This is clarified by the title that reads: “Categorisation by VMWs and LTs considering the AMM as 100% activity “ (line 232). Categories are defined in the footnote (line 233). How the AMM was calculated is described in the methods.

This table highlights the practical implications of different readings from VMWs and LTs on the same individual. Depending on measurements made, an individual may be categorized as deficient, intermediate, or normal and this may differ between VMWs and LTs with direct consequences for the choice of radical cure regimen. Table 3 highlights this discrepancy.   

  • Line 243: What statistical analysis is used?

AUTHORS: The statement is generic to explain readers how many sub-samples we retained for the overall statistical analysis.

  • Table 4: remove the second column or merge it with the first column, put the unit below the name of the column, and keep the numbers only in the table.

AUTHORS: We have revised the table as you suggested.

  • Table 5: Why are the authors divided the sample according to gender? They do not need to include the row and column total on the table and they can replace categories by their denomination

AUTHORS: The G6PD gene is located on the X-chromosome, males are either hemizygous G6PD deficient or normal. In contrast, females can also be heterozygous for the gene with intermediate activities. Accordingly, the proportion of males with intermediate activities should be lower than for females (as is the case). This distribution only becomes apparent when stratifying G6PD activities by sex. We would like to keep the “Total”, however, have replace the total proportion by row rather than column proportions to match the layout of an extended 2x2 table.

Fig. 4: What is the information that the reader needs to have here? I suggest removing the plot showing the distribution.

AUTHORS: Figure 4 is a histogram that displays the distribution of G6PD activities within the study population, overall and stratified by sex. As outlined above the distribution for males follows a bimodal form and the distribution for females a mono-modal form if the measurements are done correctly and the study population is selected at random. We have added in cut-offs to define deficient and intermediate activities based on manufacturer recommendation (red vertical lines) and based on the population specific AMM (blue vertical lines). This figure displays the differences and implications the different cut-offs have and we would like to include this figure. We found a typo in one of the headings and have replaced this.

Lines 261-262: need to come before Table 4. How are the p-values obtained?

AUTHORS: Table 4 is multi-referenced. We have referenced at the first occurrence and rest are all cross-references. P-values in the subsequent statement refers to comparison of categories therefore either Chi-squared or Fisher exact tests as outlined in methods section at line 177.

  • Lines 264-282: to prevent type I error I suggest running either a chi-square or a fisher exact test on a table that will be like this

AUTHORS: Categories that LTs used are based on a previous study as is outlined in table 4. VMWs used the same categories and categorization between LTs and VMWs is compared in table 3. The implications of applying the AMM (what the reviewer refers to as “Local”) and the manufacturer recommendations are presented in table 7. The suggested table would compare the categorization scheme from a previous study as applied by LTs to the population specific cut-off and the manufacturer recommended cut-off, but only in a subset of 114 / 1327 individuals; and we don’t think this analysis would provide additional relevant information. Apologies, we don’t think that a Chi-square test is the correct test here, as the below table would present the same results just with a different classification.

LTs

Local

Manufacturer

Normal

Intermediate

Deficient

Where each cell contains a counting number.  

Discussion

  • Lines 295-295: it is hard given the results presented to associate the discrepancy between observed count and predicted count to associate that to a lack of training since intrinsically the study does not have some checking point on wrongdoing or biased from VMW

AUTHORS: We agree, we cannot attribute their performance to our training alone, and we have not associated their biosensor performance to training, but because we did offer trainings on how to use the biosensor for G6PD measurement, we have made fair suggestions on improving further which is based on the findings from our previous study, a sub-study of the current study. We have added a statement to highlight the particular limitation that reads as follows:

The study cannot attribute the competence and output of VMWs on using biosensor based on the training alone because myriad other factors can contribute to their skills and outcome. (Line 357-359)

Reviewer 2 Report

The authors evaluated the field use of G6PD biosensor by 28 village malaria workers compared to the better trained hospital-based technicians (5 technicians; reference in this study) in Cambodia in 2021-2022 and assessed the classification scheme recommended by the manufacturer compared to that developed in the local context in Cambodia (based on the local data on adjusted male median of G6PD activity).

The background information is adequate. The methods and results are clearly described in detail. Discussion is pertinent and well written.

MAJOR COMMENTS:

none

MINOR COMMENTS:

Abstract, lines 23-26: The authors give two objectives in the main text. The second objective is just as important as the first and can be mentioned in the abstract.

Abstract: The abstract seems to be too long. Please check with the editor.

Line 72: glucose-6-phosphate dehydrogenase (G6PD)

Line 93, “a several handheld diagnostics”: Please delete “a” before “several”

Lines 93-95: several handheld diagnostics (biosensors) that provide quantitative estimates of G6PD activity within minutes have been introduced.

Line 103: Republic of Korea (ROK)

Line 110, 128: delete the extra space between “the” and “biosensor”

Line 136: coronavirus disease 2019 (COVID-19)

Line 147: subset

Line 179: units/ g haemoglobin (U/g Hb)

Lines 174-175, “since malaria may alter G6PD activity, the AMM was calculated excluding participants with confirmed P. vivax infection”: What about patients without malaria but with anaemia, which can also alter considerably G6PD activity? I think that this issue needs to be clarified in the Methods section (even if the authors explain that anaemic patients were excluded from analysis in line 200).

Line 186: “Plasmodium” (in italics) “spp.” (not in italics)

Line 189: A period after “proportions.” New sentence from “When needed,…”

Line 206: and 17/33 (51%) were females

Line 216, “Malaria RDT negative, RDT negative participants”: Malaria RDT negative participants

Table 2: “Age (years)” is not at the right place. It refers to “median (IQR)” two rows below it. The row “participant total” (1275 + 69) can be deleted from some rows since the total can be calculated by adding the number of male + female, farmer + other occupation, with or without fever. In the categories that do not add up to 1275 + 69, the legend clearly says that a subset of patients was included.  

Line 245: aparasitaemic

Line 250, “70%%”: 70%

Line 252: Table 5 and Figure 4 (instead of “Table 4 and Figure 4”)?

Figure 4: Figure 4c title: Distribution of G6PD activity in U/g Hb among (space) females

Line 262: Table 4 says that the cut-off is 6.0 U/g Hb (not “6.1 U/g Hb”).

Line 262: A period after (Table 4). A new sentence from “In two cases…”

Lines 277-278: up to one third of the study participants would have been excluded from radical cure treatment but could have been safely treated according to the local AMM

Line 304: Community health worker (CHW)

Line 322: the interpretation and corresponding treatment require

Line 329: delete the extra space between “16%” and “(61/394)”

Line 361, “G6PD categorization was trained based on previous training materials”: delete “trained”?

Line 365: delete the extra space between “compare” and “biosensor”

Ref 5: Please provide the full name of “CNM” (Centre National de Malariologie); Also “Information” (spelling)

Ref 8: Please provide the web link.

Ref 9, 22, 28: Please use the same format throughout: journal name.

Ref 12: Please use the same format throughout: article title. The reference is incomplete. Int J Mol Sci 2016, 17, 2069.

Ref 21, 26, 39: Please use the same format throughout: article title.

Ref 29: delete the second “-206331”

Ref 31 is incomplete: Elife 2021, 10, e62448.

Author Response

Date: 12th February 2023

Dear reviewers,

Thank you very much for your constructive feedback and opportunity to revise our work. We have utilized your suggestions to improve our manuscript. Please see below our responses to your specific suggestions with corresponding changes in the manuscript.

We look forward to your kind consideration.

Sincerely,

Bipin Adhikari

Reviewer #2

The authors evaluated the field use of G6PD biosensor by 28 village malaria workers compared to the better trained hospital-based technicians (5 technicians; reference in this study) in Cambodia in 2021-2022 and assessed the classification scheme recommended by the manufacturer compared to that developed in the local context in Cambodia (based on the local data on adjusted male median of G6PD activity).

The background information is adequate. The methods and results are clearly described in detail. Discussion is pertinent and well written.

 AUTHORS: Thank you for the evaluation.

MAJOR COMMENTS:

 none 

MINOR COMMENTS:

Abstract, lines 23-26: The authors give two objectives in the main text. The second objective is just as important as the first and can be mentioned in the abstract.

AUTHORS: We have revised the abstract and added both objectives and reads as follows:

The objectives of this study were to 1. compare the G6PD activity readings using biosensor by Village malaria workers (VMWs) and hospital-based laboratory technicians (LTs); and 2. compare the G6PD deficiency categorization recommended by the biosensor manufacturer with categories derived from a locally estimated adjusted male median (AMM) in Kravanh district, Cambodia. (Line: 22-26)

Abstract: The abstract seems to be too long. Please check with the editor.

AUTHORS: We have revised and shortened the abstract.

Line 72: glucose-6-phosphate dehydrogenase (G6PD)

AUTHORS: Thank you for the suggestion. We have explained the acronym in the title and abstract (line: 1 and 19-20).  

Line 93, “a several handheld diagnostics”: Please delete “a” before “several”

AUTHORS: Revised as suggested.

Lines 93-95: several handheld diagnostics (biosensors) that provide quantitative estimates of G6PD activity within minutes have been introduced.

AUTHORS: Revised as suggested.

Line 103: Republic of Korea (ROK)

 AUTHORS: Revised as suggested.

Line 110, 128: delete the extra space between “the” and “biosensor”

AUTHORS: Revised as suggested.

Line 136: coronavirus disease 2019 (COVID-19)

AUTHORS: Revised as suggested.

Line 147: subset

AUTHORS: Revised as suggested.

Line 179: units/ g haemoglobin (U/g Hb)

AUTHORS: Revised as suggested.

Lines 174-175, “since malaria may alter G6PD activity, the AMM was calculated excluding participants with confirmed P. vivax infection”: What about patients without malaria but with anaemia, which can also alter considerably G6PD activity? I think that this issue needs to be clarified in the Methods section (even if the authors explain that anaemic patients were excluded from analysis in line 200).

AUTHORS: We have revised the respective section which now reads: “The Biosensor measures Hb levels within a range of 4g/dL to 25g/dL, however, G6PD readings can only be considered reliable if Hb readings are above 7g/dL based on the manufacturer’s recommendation. Further on G6PD activities in anaemic individuals can be artificially increased [30]. G6PD readings with a corresponding Hb level below 7g/dL or above 25g/dL were therefore excluded from the analysis.” Lines 157 – 161). The newly added reference [30] is: Satyagraha AW, Sadhewa A, Baramuli V, Elvira R, Ridenour C, Elyazar I, Noviyanti R, Coutrier FN, Harahap AR, Baird JK: G6PD deficiency at Sumba in Eastern Indonesia is prevalent, diverse and severe: implications for primaquine therapy against relapsing Vivax malaria. PLoS Negl Trop Dis 2015, 9:e0003602.

Line 186: “Plasmodium” (in italics) “spp.” (not in italics)

AUTHORS: Revised as suggested.

Line 189: A period after “proportions.” New sentence from “When needed,…”

AUTHORS: Revised as suggested.

Line 206: and 17/33 (51%) were females

AUTHORS: Revised as suggested.

Line 216, “Malaria RDT negative, RDT negative participants”: Malaria RDT negative participants

AUTHORS: Revised as suggested.

Table 2: “Age (years)” is not at the right place. It refers to “median (IQR)” two rows below it. The row “participant total” (1275 + 69) can be deleted from some rows since the total can be calculated by adding the number of male + female, farmer + other occupation, with or without fever. In the categories that do not add up to 1275 + 69, the legend clearly says that a subset of patients was included.  

 AUTHORS: Revised as suggested. 

Line 245: aparasitaemic

AUTHORS: Revised as suggested. 

Line 250, “70%%”: 70%

AUTHORS: Revised as suggested. 

Line 252: Table 5 and Figure 4 (instead of “Table 4 and Figure 4”)?

AUTHORS: Revised as suggested.  

Figure 4: Figure 4c title: Distribution of G6PD activity in U/g Hb among (space) females

AUTHORS: Revised as suggested.

Line 262: Table 4 says that the cut-off is 6.0 U/g Hb (not “6.1 U/g Hb”).

AUTHORS: Revised as suggested.  

Line 262: A period after (Table 4). A new sentence from “In two cases…”

AUTHORS: Revised as suggested.  

Lines 277-278: up to one third of the study participants would have been excluded from radical cure treatment but could have been safely treated according to the local AMM

AUTHORS: Revised as suggested.  

Line 304: Community health worker (CHW)

AUTHORS: Revised as suggested.   

Line 322: the interpretation and corresponding treatment require

AUTHORS: Revised as suggested.   

Line 329: delete the extra space between “16%” and “(61/394)”

AUTHORS: Revised as suggested.   

Line 361, “G6PD categorization was trained based on previous training materials”: delete “trained”?

AUTHORS: Revised as suggested.    

Line 365: delete the extra space between “compare” and “biosensor”

AUTHORS: Revised as suggested.   

Ref 5: Please provide the full name of “CNM” (Centre National de Malariologie); Also “Information” (spelling)

AUTHORS: Revised as suggested.   

Ref 8: Please provide the web link.

AUTHORS: Revised as suggested.   

Ref 9, 22, 28: Please use the same format throughout: journal name.

AUTHORS: Revised as suggested.   

Ref 12: Please use the same format throughout: article title. The reference is incomplete. Int J Mol Sci 2016, 17, 2069.

AUTHORS: Revised as suggested.   

Ref 21, 26, 39: Please use the same format throughout: article title.

AUTHORS: Revised as suggested.   

Ref 29: delete the second “-206331”

AUTHORS: Revised as suggested.   

Ref 31 is incomplete: Elife 2021, 10, e62448.

AUTHORS: Revised as suggested.   

Round 2

Reviewer 1 Report

Remove the blue background and the bounding box of the figures. 

Author Response

Date: 23rd February 2023

Thank you very much for your kind consideration and a suggestion. We have made changes based on your suggestion.  

Sincerely,

Bipin Adhikari

Reviewer #1 

Remove the blue background and the bounding box of the figures. 

AUTHORS: We have revised the figures as suggested.